# Controlling Gut Inflammation by Restoring Anti-Inflammatory Pathways in Inflammatory Bowel Disease

**DOI:** 10.3390/cells8050397

**Published:** 2019-04-30

**Authors:** Paolo Giuffrida, Sara Cococcia, Mariangela Delliponti, Marco Vincenzo Lenti, Antonio Di Sabatino

**Affiliations:** First Department of Internal Medicine, University of Pavia and Fondazione IRCCS Policlinico San Matteo, 27100 Pavia, Italy; paolo.giuffrida01@universitadipavia.it (P.G.); sara.cococcia@gmail.com (S.C.); m.delliponti@smatteo.pv.it (M.D.); marco.lenti@unipv.it (M.V.L.)

**Keywords:** Crohn’s disease, granulocyte macrophage colony-stimulating factor, interleukin-10, mesenchymal stem cells, regulatory T cells, tolerogenic dendritic cells, transforming growth factor-β, ulcerative colitis

## Abstract

Inflammatory bowel disease (IBD) is caused by a dysregulated immune response against normal components of the intestinal microflora combined with defective functioning of anti-inflammatory pathways. Currently, all therapies approved for IBD manipulate the immune system by inhibiting pro-inflammatory mechanisms, such as tumor necrosis factor-α, gut-homing α_4_β_7_ integrin, interleukin-12/interleukin-23, and Janus kinases. However, some IBD patients are non-responders to these drugs, which are also associated with serious side effects. Thus, it has been hypothesized that therapies aimed at restoring anti-inflammatory signals, by exploiting the tolerogenic potential of cytokines (interleukin-10, transforming growth factor-β, granulocyte macrophage colony-stimulating factor), immune cells (regulatory T cells, tolerogenic dendritic cells), or mesenchymal stem cells, might offer promising results in terms of clinical efficacy with fewer side effects. In this review, we provide new insights into putative novel treatments aimed at restoring anti-inflammatory signaling pathways in IBD.

## 1. Introduction

Pathogenic mechanisms underlying inflammatory bowel disease (IBD) rely on an abnormal immune response against the normal microbiota [1]. Hyperactivation of pro-inflammatory pathways is associated with defective counter-regulatory signaling in inflamed intestinal tissues of patients with IBD [2]. Our understanding of the mucosal immunology in IBD has been improved notably over the last few years. This notwithstanding, all currently available and approved therapies for IBD patients block the aberrant immunity only by dampening pro-inflammatory pathways, i.e., tumor necrosis factor (TNF)-α, gut-homing α_4_β_7_ integrin, interleukin (IL)-12/IL-23, and Janus kinases [3]. However, some IBD patients are not responsive to the aforementioned drugs [4], which also induce serious side effects, including opportunistic infections and malignancies [5]. Thus, the clinical management of IBD patients through completely safe and effective therapies is still an unmet need. As anti-inflammatory pathways—such as the cytokines IL-10, transforming growth factor (TGF)-β, and granulocyte macrophage colony-stimulating factor (GM-CSF), regulatory T cells (Tregs), tolerogenic dendritic cells (DCs), and mesenchymal stem cells (MSCs)—miss immune response in IBD (Figure 1), treatments aimed at bolstering them have been investigated for years. In particular, some molecules targeting anti-inflammatory mechanisms showed preliminary promising results (Table 1) [6]. In this review, we analyze new insights into putative novel treatments by restoring the anti-inflammatory response in IBD.

## 2. Cytokine-Based Anti-Inflammatory Therapy

### 2.1. Interleukin-10

IL-10 is a key immunoregulatory cytokine that can be expressed by virtually any innate and adaptive immune cell, including macrophages, natural killer cells, eosinophils, neutrophils, and B- and T-lymphocytes [24,25]. Although it was generally considered as an anti-inflammatory cytokine that limits and terminates immune responses [26], IL-10 has pleiotropic effects, including both immunostimulatory properties against infectious and noninfectious noxious agents, and immunosuppressive properties, especially toward eosinophils and allergic disorders [27]. Patients suffering from IBD have high levels of IL-10, especially during a flare [28], suggesting that this cytokine could represent a target for IBD therapy. In addition, children affected by rare mutations in the IL-10 receptor, which make IL-10 signaling defective, are more likely to suffer from very early-onset IBD, with an onset occurring within the first six months after birth [29,30]. Starting from these premises, several studies evaluated both the safety and efficacy of IL-10-based therapies, demonstrating their safety, tolerability, and clinical efficacy over a variable treatment period, from one week to one month. [31,32,33]. However, these preliminary findings were not completely confirmed by further studies. A randomized, double-blind, placebo-controlled, multicenter trial conducted in seven different European countries showed that recombinant human IL-10 (Tenovil) fails to prevent postoperative recurrence of CD in patients treated within two weeks from resection [7]. In addition, safety concerns are emerging, as IL-10 is associated with relevant side effects due to its pro-inflammatory properties, through the activation of interferon (IFN)-γ [32,34,35]. Furthermore, treatment with IL-10 has been associated with worsening of anemia, which is frequently observed in patients suffering from chronic inflammatory disorders [36]. The best delivery method for IL-10 therapy is still debated in terms of reduction of side effects and improvement of clinical response. In a phase 1 trial, IL-10 was administered through genetically-modified bacteria [8]. The use of genetically-modified *Lactobacillus lactis* (LL-Thy12) for direct mucosal delivery proved to have a significantly better safety profile, reducing the frequency of the side effects commonly registered in other trials [8]. Up to date, systemic IL-10 therapy has been disappointing, as no clinical trial has demonstrated a clear role of IL-10 in treating IBD. Nonetheless, the novel immunocytokine F8-IL-10 (Dekavil), which use a targeting antibody linked to IL-10 that is now being evaluated in a phase 2 clinical trial in patients with rheumatoid arthritis, may pave the way for its use in IBD patients [37]. Further studies with the aim of defining the efficacy and safety of IL-10-based drugs are required to better understand the potential clinical efficacy of this class of treatments.

### 2.2. Transforming Growth Factor-β

The expression of the anti-inflammatory cytokine TGF-β is unexpectedly high in the inflamed mucosa of IBD patients in comparison to control gut [38]. However, TGF-β signaling is deficient in IBD mucosa, as shown by low amounts of its transcriptional proteins phosphorylated Smad2/3 therein [39]. This is due to enhanced levels of the inhibitory protein Smad7, which blocks Smad2 and Smad3 phosphorylation by binding to TGF-β receptor I [39]. The knockdown of Smad7 through a specific antisense oligonucleotide restores TGF-β signaling, as suggested by an increase in phosphorylated Smad3, thus leading to a strong reduction of the pro-inflammatory cytokines TNF-α and IFN-γ from CD lamina propria mononuclear cells [39]. Oral administration of a specific Smad7 antisense oligonucleotide dampens inflammation by upregulating phosphorylated Smad3 in experimental murine models with trinitrobenzene sulfonic acid (TNBS)- and oxazolone-induced colitis, which are both characterized by mucosal increase in TGF-β and Smad7 and reproduce CD and UC, respectively [40]. These data encouraged the development of the pharmaceutical compound Mongersen, formerly known as GED0301, as a therapeutic strategy in IBD [9]. Mongersen is a single-stranded DNA oligonucleotide matching the region 107–128 of the human Smad7 DNA sequence and contains two CpG motifs chemically modified in order to escape immunostimulatory activity. Mongersen is administered orally, and its active compound is principally released in the terminal ileum and right colon, the most frequent locations affected in CD. In a phase 1, open-label, single-center, dose-escalation study, Mongersen was administered once daily for seven consecutive days to 15 patients with moderate-to-severe, steroid-dependent/resistant CD [9]. Mongersen induced clinical response in all patients and clinical remission in 13/15 (86%) patients at Day 28 and was safe with no drug-related adverse events [9]. In addition, Mongersen was associated with a significant decrease in circulating IFN-γ^+^ and IL-17A^+^ T cells expressing the gut-homing protein CCR9 [9]. As TGF-β is a pro-fibrogenic cytokine [41,42], patients recruited in the phase 1 trial were followed-up at Day 180 with a small intestine contrast ultrasonography showing no strictures in each patient [43]. Moreover, at Day 180, no patients had a change in the serum levels of the tissue inhibitor of matrix metalloproteinases-1, basic fibroblast growth factor and YKL-40 [43], which have been proposed as serum biomarkers for intestinal fibrosis [44]. A further study demonstrated that Mongersen limits fibrosis in murine models with chronic TNBS-induced colonic fibrosis [45]. In a phase 2, multicenter, double-blind, placebo-controlled trial, 166 patients with moderate-to-severe, steroid-dependent/resistant CD were randomized in a 1:1:1:1 ratio to receive one of three doses of Mongersen (10, 40, or 160 mg per day) or placebo daily for two weeks [10]. The primary endpoint, i.e., clinical remission at Day 15 maintained for a further two weeks, was achieved by 65% and 55% of patients belonging to the 160-mg and 40-mg groups, respectively, which was much higher than the percentage of patients receiving 10 mg of Mongersen per day (12%) or placebo (10%) [10]. The safety profile of Mongersen was also confirmed in this study [10]. A post hoc analysis of this phase 2 study demonstrated that CD patients with baseline CDAI ≤ 260 had significantly increased remission rates with 40 and 160 mg of Mongersen per day, whereas in CD patients with baseline Crohn’s Disease Activity Index (CDAI) >260, remission rates were statistically higher in the 160-mg group [46]. A phase 1b multicenter, double-blind, placebo-controlled study then demonstrated endoscopic improvement in 37% of CD patients at Week 12 [11]. This notwithstanding, a phase 3, multicenter, double-blind, placebo-controlled trial in CD was terminated in advance due to no efficacy of Mongersen according to an interim analysis [12]. A phase 2, open-label study was recently completed in UC patients, and results are expected soon [13].

### 2.3. Granulocyte Macrophage Colony-Stimulating Factor

The GM-CSF is a glycoprotein that enhances the function of mature white cells, including neutrophils, monocytes, and macrophages, and stimulates the expansion and differentiation of hemopoietic progenitors, acting as a potent stimulant of the innate immunity [14,47,48]. There are studies suggesting that intestinal inflammation in CD may be the consequence of a primary deficiency of innate immunity [49]. Indeed, a CD-like phenotype may be found in animal models in the presence of impaired innate immunity or mucosal barrier [50,51,52,53] and in rare human immunodeficiency diseases [54,55,56,57,58,59,60,61,62,63] characterized by a selective deficiency, either quantitative or qualitative, of the neutrophil function or of innate cells. This supports the hypothesis that a defect of the innate immunity in the gastrointestinal mucosa may play a relevant role in the pathogenesis of CD. Based on these observations, the safety and effectiveness of recombinant human GM-CSF was evaluated in CD by different studies [14,15]. An eight-week, open-label, dose-escalating study (4, 6, and 8 μg/kg daily), enrolling 15 patients with a CDAI score between 221 and 474, reported an 80% rate of clinical response (decrease of CDAI greater than 70 points) and a 53% rate of clinical remission, defined as CDAI below 150. After therapy discontinuation, sustained response was extremely variable, with a median of eight weeks before any therapeutic intervention became necessary. The main reported side effect was medullary bone pain, experienced by ten out of 15 patients. The effect on the circulating neutrophils and eosinophils was found to be dose-dependent, and no significant difference was found between responders and non-responders. [14] A randomized, placebo-controlled, phase 2 trial [15], enrolling 124 patients with moderate-to-severe CD receiving either 6 μg/kg of GM-CSF (Sargramostim) daily or placebo subcutaneously for 57 days, failed in achieving the primary outcome (i.e., a reduction in CDAI score of 70 or more). Nevertheless, a significantly greater number of patients in the Sargramostim arm had both a CDAI reduction of at least 100 points at the end of treatment (48% vs. 26%, *p* = 0.01) and achieved remission (40% vs. 19%, *p* = 0.01), compared to the placebo arm. In addition, the rate of response was significantly higher at all the considered time points, including 30-day follow-up (*p* = 0.03), as well as the rate of remission at the end of the 30-day follow-up (*p* = 0.02), and a significant improvement of the quality of life was demonstrated comparing the Sargramostim group to the placebo one at all time points, but especially at 30-day follow-up (*p* = 0.006) [15]. Due to the poor evidence derived from the randomized trials conducted with Sargramostim, further research is needed to verify the efficacy of GM-CSF in CD. At present, there are no active trials investigating this molecule in IBD patients.

## 3. Cell-Based Anti-Inflammatory Therapy

### 3.1. Regulatory T Cells

Tregs encompass a heterogeneous group of T cells that are involved in maintaining the peripheral tolerance to both self- and external antigens, thus downregulating unnecessary inflammatory processes [64,65]. Regardless of their origin, Tregs have a high affinity with IL-2 receptor α-chain (CD45) and Foxp3, which is a key factor for the maintenance of their suppressive phenotype [66,67]. Type 1 Tregs differentiate peripherally and are activated by an antigen-specific pathway, regulating the response of other T cells (naive and memory) through the secretion of IL-10 [53,67]. Animal models showed that ovalbumin-specific type 1 Tregs (ova-Tregs), generated in vitro from CD4 T cells, can dampen the inflammatory response and, as a result, can control colitis in mice [67,68,69]. Ova-Tregs are recruited at the inflamed sites, where they are triggered by antigen-presenting cells through the ovalbumin antigen in combination with soluble factors (IL-10, granzyme B) and membrane-bound molecules (CD39, CTLA-4, and GITR) [69]. So far, only one study, the CATS1 (Crohn’s And Tregs Cells Study) trial, has evaluated the safety and efficacy of Treg therapy in IBD [16]. In this 12-week, open-label, multicenter, single-injection, escalating-dose, phase 1/2a study, 20 patients with refractory CD were enrolled and treated with a single infusion of ova-Tregs isolated from their own peripheral blood mononuclear cells. Patients received one out of four different doses (10^6^, 10^7^, 10^8^, 10^9^ autologous ova-Tregs) and were followed-up for 12 weeks. The results showed that 40% of patients responded at Weeks 5 and 8 with a CDAI score reduction of 100 points or more and a significantly higher reduction of the same score in those receiving doses of 10^6^ cells at both time points. Nevertheless, the benefit did not last over time, as 12 weeks after infusion, the CDAI score was comparable to baseline [16]. In addition, this study did not assess the homing capacity of ova-Tregs, nor the expression of suppressive molecules on Tregs. Actually, the suppressive phenotype of ova-Tregs might be unstable. A further study on CD patients’ blood cells expanded in vitro compared CD45RA-activated Tregs and CD45RA^+^ resting Tregs and showed that the latter have an epigenetically-stable *FOXP3* locus and do not switch into a Th17 phenotype in vitro [70]. Furthermore, CD45RA^+^ Tregs express a high level of α_4_β_7_ integrin, CD62L, and CC motif receptor 7 and have been shown to home to the human small bowel in a xenotransplant model of severe combined immunodeficiency [70]. Another key aspect noticed is that inflammation of the lamina propria in active CD mucosa and in mesenteric lymph nodes is healed by CD45RA^+^ Tregs [70]. At present, there is an ongoing study (TRIBUTE trial), which aims to treat active CD patients with in vitro expanded Tregs [17].

### 3.2. Tolerogenic Dendritic Cells

DCs migrate from peripheral tissues to T cell areas of secondary lymphoid organs, including the gut, where they act as antigen-presenting cells [71]. Upon stimulation of TGF-β and retinoic acid from intestinal epithelial cells, mucosal DCs express CD103 and become tolerogenic, so enabling not only the differentiation of Tregs, but also the inhibition of Th1 and Th17 cell development [72]. Resident DCs protect the gut during the early phases of adoptive transfer colitis models [73], by promoting α_4_β_7_-integrin^+^ and CCR9^+^ Tregs through TGF-β and retinoic acid [74,75,76,77]. Conversely, in murine models of advanced colitis mediated by adoptive transfer, CD103^+^ DCs display low production of TGF-β and reduced activity of the *ALDH1A2* gene, one of the enzymes involved in retinoic acid synthesis together with *ALDH1A1* and *ALDH1A3*, thus leading to a limited amount of Foxp3^+^ Tregs and an increase in IFN-γ-producing T cells [78]. Likewise, CD103^+^ DCs isolated from patients with CD do not express the *ALDH1A1* enzyme [79].

A phase 1, single-center, sequential-cohort, dose-range study investigated the safety and tolerability of autologous tolerogenic DCs, derived from monocytes by leukapheresis and administered by ultrasound-guided intraperitoneal injection in nine patients with refractory moderate-to-severe CD [18,80]. The study included six sequential cohorts with six different regimens on the basis of the amount of administered cells (2 × 10^6^ DCs/mL, 5 × 10^6^ DCs/mL, 10 × 10^6^ DCs/mL) and the number of doses (one or three every two weeks) [18]. Amongst the nine CD patients, one obtained clinical remission and two clinical response, whereas CDAI was reduced from 274 (baseline) to 222 (at Week 12) in all the patients [18]. In parallel, a marked endoscopic improvement was shown in three out of nine patients [18]. No adverse effect was observed in any patient during treatment and one-year-long follow-up [18]. The percentage of peripheral blood Treg was significantly increased at Week 12 in comparison to that of the baseline, whereas no difference was found between baseline and Week 12 in terms of Th1 and Th17 cells [18]. In addition, peripheral blood mononuclear cells stimulated with CD3 antibody produced a lesser amount of IFN-ã at week 4 [18]. These findings suggest that tolerogenic DC administration dampens the pro-inflammatory response by enhancing Tregs in CD. Currently, another clinical trial is ongoing in order to assess the safety and the clinical efficacy of autologous tolerogenic DCs administrated into visible lesions by endoscopy in refractory CD [19].

### 3.3. Mesenchymal Stem Cells

Amongst stem cells, MSCs are the best candidate as therapy in IBD due to their migration to inflamed intestinal tract, where they exert a regenerative function and the absence of immunogenicity [81]. In addition, MSCs display immunoregulatory effects through a direct intercellular contact and release of soluble factors [82]. In particular, MSCs have been shown to inhibit T cell proliferation [83], dampen cytotoxicity from T cells [84], and promote the generation of Tregs [85] and IL-10-producing Th1 cells [86]. In a single-center study, autologous bone marrow-derived MSCs were administered into perianal fistulas of ten CD patients [20]. Amongst them, seven obtained a complete closure and three a partial closure of fistula with a concomitant healing in the rectal mucosa [20]. The percentage of mucosal and peripheral Tregs increased along the treatment and remained stable up to the end of the 12 month-long follow-up [20]. Moreover, MSCs were able to affect mucosal T cell apoptotic rate [20]. The likelihood of fistula relapse-free survival in these ten CD patients was 88% at one year, 50% at two years, and 37% during the following four years, whereas the cumulative probabilities of surgery- and medical-free survival were 100% and 88% at one year, 75% and 25% at two, three, and four years, and 63% and 25% at five and six years, respectively [87]. No adverse events have been reported in the six year-long follow-up [20,87]. In a phase 3 randomized, double-blind, parallel-group, placebo-controlled study, 212 CD patients with complex perianal fistulas were randomly assigned 1:1 to allogeneic adipose-derived MSCs, also termed as Cx601, and to placebo [21]. A higher proportion of patients given Cx601 achieved combined remission at Week 24 (50%) in comparison to the placebo group (34%) in the intention-to-treat [21]. At Week 52, an increased percentage of patients treated with Cx601 achieved combined remission (56.3%) compared to the placebo arm (38.6%) [88]. No significant difference in terms of side effects was found between Cx601 and placebo [88]. Cx601 received a positive opinion from the Committee for Medicinal Products for Human Use on 15 December 2017.

## 4. Other Immunotherapies

### 4.1. Autologous Colonic Proteins

Immunotherapy represents an intriguing therapeutic option that has the aim of restoring the physiological T cell tolerance towards specific antigens, while leaving the rest of the immune system unaffected. Antigen- and epitope-based immunotherapy has rapidly developed over the last decade, especially for oncologic and allergic disorders, in which immunodominant disease triggers are increasingly recognized, and therefore can be targeted [89]. The development of immunotherapy is more complex for autoimmune and immune-mediated diseases, given that many different antigens play a role in disease onset, as in the case of CD. However, the oral route of administration could potentially be the most feasible and effective in CD, considering that oral exposure to external antigens is a physiological process that triggers gut immune cell activation and tolerance [90]. Starting from these premises, a phase 2 study assessing the safety and efficacy of oral administration of Alequel, a mixture of proteins derived from patients’ bowel, was completed in 2014 [23,91]. In the phase 1 trial, no safety signals emerged, and seven out of ten patients achieved clinical remission within the study period [22]. The phase 2, randomized, double-blind, placebo-controlled trial confirmed the good safety profile over a 27-week period, but no statistical significance was noticed between the drug and placebo groups for both clinical response and remission, using an intention to treat analysis [23]. Notably, in the drug-treated cohort who achieved remission, a decreased number of subject-specific, antigen-directed, IFN-γ spot-forming colonies was noticed, as well as an increased number of peripheral blood NK cells and an increased CD4+/CD8+ T cell ratio during drug administration. Unfortunately, the small sample size does not allow drawing firm conclusions, and larger studies are needed to assess the efficacy of this approach in IBD.

### 4.2. Otelixizumab

Otelixizumab is a chimeric/humanized Fc-engineered monoclonal antibody against CD3/ε showing encouraging results in patients with new or recent onset of diabetes mellitus type 1 [92,93]. However, lower doses of otelixizumab did not achieve the primary endpoints of the following phase 3 trials in new-onset diabetes mellitus type 1 [94,95]. Otelixizumab up-regulated the anti-inflammatory cytokine IL-10 production from IBD explants and lamina propria mononuclear cells. IL-10 plays a pivotal role in the otelixizumab-related phosphoprotein reduction, as suggested by the retention of high phosphorylation status in IBD biopsies after stimulating them in the presence of both an IL-10 neutralizing antibody and otelixizumab itself [96]. Of note, in ex vivo experiments, otelixizumab also reduced the expression of phosphoproteins, in particular not only those related to TCR signaling, but also other ones not associated with immune cells [96]. Briefly, otelixizumab seems to induce IL-10 production through the expansion of Tregs [96]. This is in agreement with the increase in IL-10 release from intestinal Tregs in humanized mice treated with teplizumab, a humanized Fc receptor nonbinding monoclonal antibody to CD3 [97]. Moreover, it has been demonstrated that otelixizumab downregulated a broad range of pro-inflammatory cytokines and chemokines released from IBD biopsies in the same experimental setting [96]. However, additional studies are clearly needed in order to understand clearly the mechanisms of otelixizumab in IBD. Despite the powerful anti-inflammatory effects caused by otelixizumab in IBD samples both in vitro and ex vivo experiments, no clinical trial using this drug has been hitherto designed in IBD patients.

## 5. Conclusions

In addition to the current therapeutic strategies blocking pro-inflammatory cytokines, lymphocyte gut homing, and Janus kinases, several novel treatments aimed at bolstering the immunoregulatory pathways have been proposed for the treatment of IBD patients (Table 1). Cytokine-based therapies—Tenovil, LL-Thy12, Mongersen, and Sargramostim—and autologous colonic proteins showed no clear evidence of efficacy and safety in all the trials conducted so far. On the other hand, cell-based anti-inflammatory therapies, i.e., Tregs, tolerogenic DCs, and MSCs, provided promising results in phase 1/2a, 1, and 3 trials, respectively. Studies investigating otelixizumab in IBD are still lacking, although the powerful immunoregulatory effect caused by this drug in in vitro/ex vivo experiments seems to be encouraging. In conclusion, we are still far away from the use of drugs modulating anti-inflammatory response in IBD in daily clinical practice. Amongst them, cell-based therapies might enter the clinical management of IBD patients, if their safety profile and clinical/endoscopic effectiveness are confirmed in the ongoing/following trials. In the future, anti-inflammatory drugs look to be efficacious as single agents due their intrinsic action aimed at dampening pro-inflammatory response in IBD. Conversely, a combined treatment by restoring anti-inflammatory mechanisms and inhibiting pro-inflammatory pathways might be indicated only in selected patients with refractory IBD and an aggressive behavior according to treat-to-target strategy and personalized medicine.

## Figures and Tables

**Figure 1 cells-08-00397-f001:**
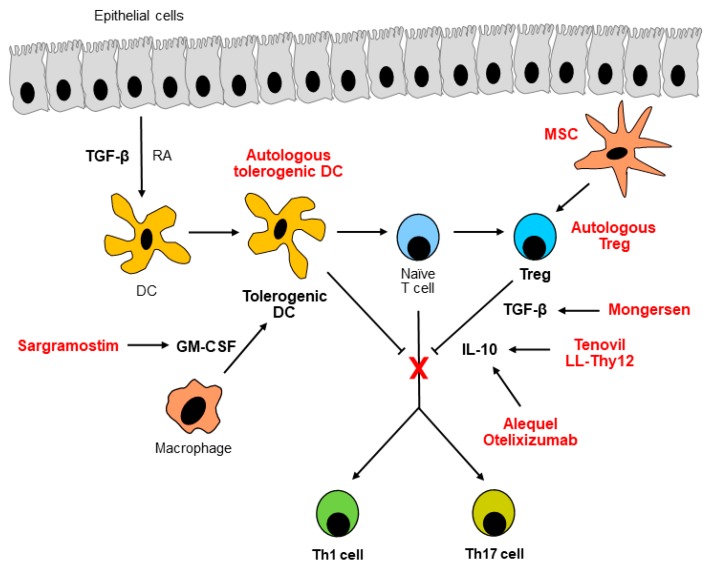
Schematic representation of the anti-inflammatory mechanisms dampening the inflammation in the gut. Upon stimulation of transforming growth factor (TGF)-β and retinoic acid (RA) from epithelial cells and granulocyte macrophage colony-stimulating factor (GM-CSF) from macrophages, dendritic cells (DCs) become tolerogenic, thus promoting the differentiation of regulatory T cells (Tregs) from naive T cells in healthy gut. Sargramostim and autologous tolerogenic DCs resume the function of GM-CSF and tolerogenic DCs themselves, respectively. In addition to tolerogenic DCs, Tregs inhibit development of T helper 1 (Th1) and Th17 cells by producing TGF-β and interleukin (IL)-10 in healthy gut. Autologous Tregs reinstate Treg function. Mongersen reactivates TGF-β signaling, whereas Tenovil, IL-10 administered through genetically-modified *Lactobacillus lactis* (LL-Thy12), Alequel, and otelixizumab restore IL-10. Mesenchymal stem cells (MSCs) induce an increase in the number of Tregs.

**Table 1 cells-08-00397-t001:** Molecules restoring anti-inflammatory signals in inflammatory bowel disease.

Publication Year	First Author	Drug Used	Drug Type	Route	IBD	N	Placebo Group	Intervention Group(s)	Primary Endpoint	Statistical Benefit (*p*-value)	Clinical Benefit or Harm
2001	Colombel [7]	Tenovil	Rh cytokine	sc	CD	65	22	43	Safety and tolerance within 2 weeks of the first ileal or ileocolonic resection	No (NS)	None
2006	Braat [8]	LL-Thy12	Genetically modified bacteria	Oral	CD	10	NA	10	Safety	NA	Benefit
2012	Monteleone [9]	Mongersen	Oligonucleotide	Oral	CD	15	NA	15	Safety and tolerance	NA	Benefit
2015	Monteleone [10]	Mongersen	Oligonucleotide	Oral	CD	166	42	124	Clinical remission at Day 15	Yes (<0.0001)	Benefit
2018	Feagan [11]	Mongersen	Oligonucleotide	Oral	CD	63	NA	63	Effect on endoscopic CD measures	NA	Benefit
NA	[12]	Mongersen	Oligonucleotide	Oral	CD	701	UKN	UKN	Clinical remission at Week 12	No	None
NA	[13]	Mongersen	Oligonucleotide	Oral	UC	41	NA	41	Clinical remission at Week 8	UKN	UKN
2002	Dieckgraefe [14]	Sargramostim	Rh cytokine	sc	CD	15	NA	15	Safety and effectiveness	NA	Benefit
2005	Korzenik [15]	Sargramostim	Rh cytokine	sc	CD	124	43	81	Clinical response at Day 57	No (= 0.28)	None
2012	Desreumaux [16]	ova-Tregs	Autologous cells	iv	CD	20	NA	20	Safety and tolerability	NA	Benefit
NA	[17]	In vitro expanded Tregs	Autologous cells	iv	CD	UKN	UKN	UKN	Rate of dose limiting toxicities and determination of maximum tolerated dose	UKN	UKN
2015	Jauregui-Amezaga [18]	Tolerogenic DCs	Autologous cells	ip	CD	9	NA	9	Safety and tolerability	NA	Benefit
NA	[19]	Tolerogenic DCs	Autologous cells	il	CD	UKN	NA	UKN	Number of adverse events and proportion of patients with clinical response	NA	UKN
2011	Ciccocioppo [20]	MSCs	Autologous cells	if	CD	10	NA	10	Safety and efficacy	NA	Benefit
2016	Panes [21]	Cx601	Allogeneic cells	if	CD	212	107	105	Combined remission at Week 24	Yes (<0.05)	Benefit
2005	Israeli [22]	Alequel	ACP	Oral	CD	10	NA	10	Safety and tolerability	NA	Benefit
2006	Margalit [23]	Alequel	ACP	Oral	CD	31	15	16	Clinical response and remission	No (NS)	None

ACP, autologous colonic protein; CD, Crohn’s disease; DC, dendritic cell; if, intrafistular; il, intralesional; ip, intraperitoneal; iv, intravenous; LL-Thy12, genetically-modified *Lactobacillus lactis*; MSC, mesenchymal stem cell; NA, not applicable; NS, not significant; Rh, recombinant human; sc, subcutaneous; Treg, regulatory T cell; UC, ulcerative colitis, UKN, unknown.

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
