# Peer review of "Controlling Gut Inflammation by Restoring Anti-Inflammatory Pathways in Inflammatory Bowel Disease"

_cells, 2019, doi:10.3390/cells8050397_

Round 1
Reviewer 1 Report
This article by Giuffrida et al. provided a through review on the current status of the use of anti-inflammatory pathways in the treatment of IBD. This reviewer likes this short review a lot, but it would be nice if the authors can discuss on whether a combined therapy is possible by inhibiting pro-inflammatory and restoring anti-inflammation in IBD treatment.
Author Response
We thank the Reviewer for his/her helpful comment and for his/her appreciation of the review. We have attempted to answer the point raised by discussing on it in the revised manuscript (page 9, line 308).
Reviewer 2 Report
Giuffrida et al. is a summarized review article about the current conservative interventions in IBD management. To date, all therapies approved for IBD treatment manipulate the immune system by inhibiting pro-inflammatory mechanisms, such as tumor necrosis factor-α, gut-homing α4β7 integrin, interleukin-12/interleukin-23 and Janus kinases. However, some IBD patients are non-responder to these drugs, which are also associated to serious side effects. Thus, it has been hypothesized that therapies aimed at restoring anti-inflammatory signals, by exploiting the tolerogenic potential of cytokines (interleukin-10, transforming growth factor-β, granulocyte macrophage colony-stimulating factor), immune cells (regulatory T cells, tolerogenic dendritic cells) or mesenchymal stem cells, might offer promising results in terms of clinical efficacy with fewer side effects. In this review, the authors provide new insights into putative novel treatments aimed at restoring anti-inflammatory signaling pathways in IBD.
They demonstrate in Table 1, that Cytokine-based therapies i.e. Tenovil, LL-Thy12, Mongersen and Sargramostim- and autologous colonic proteins showed no clear evidence of efficacy and safety in all the trials conducted so far. On the other hand, cell-based anti-inflammatory therapies –i.e. Tregs and tolerogenic DCs- provided promising results in phase I/IIa and I trials, respectively. Studies investigating otelixizumab in IBD are still lacking, although the powerful immuregulatory effect caused by this drug in in vitro/ex vivo experiments seems to be very promising. The authors concluded by admitting the fact that we are still far away on the use of drugs modulating anti-inflammatory response in IBD in daily IBD clinical practice. I find this review informative and of potential educational values. It is well written, presented extrovertly well and easy to follow. It is publishable unaltered in its current version.
Author Response
We thank the Reviewer for his/her appreciation of the review.